# A Systematic Review of the Diagnosis and Treatment of Non-Typhoid Salmonella Spondylodiscitis in Immunocompetent Children

**DOI:** 10.3390/children9121852

**Published:** 2022-11-29

**Authors:** Galateia Katzouraki, Elias S. Vasiliadis, Vasileios Marougklianis, Dimitrios Stergios Evangelopoulos, Spyros G. Pneumaticos

**Affiliations:** 3rd Department of Orthopaedics, School of Medicine, National and Kapodistrian University of Athens, KAT Hospital, 14561 Athens, Greece

**Keywords:** spondylodiscitis, discitis, non-typhoid *Salmonella*, children

## Abstract

The aim of this systematic review is to distinguish the clinical features of immunocompetent children with non-typhoid *Salmonella* spondylodiscitis and summarize the diagnosis, diagnostic tools, and treatment methods to guide clinicians. The review was conducted according to the preferred PRISMA guidelines. We conducted a literature search in the PubMed, Embase, and Cochrane Library databases. Article screening, data extraction, and study evaluation were performed by two independent reviewers. A total of 20 articles, published between 1977 and 2020, were selected, which included 21 patients with average age of 12.76 years (range, 2–18) without comorbidities; in total, 19% of the patients had positive blood cultures for non-typhoid *Salmonella*, and 80.9% underwent either CT-guided or open biopsy, which were positive for NTS. All infections were monomicrobial, and 11 different serotypes of non-typhoid *Salmonella* were identified. Analyzing the reviewed cases, 52.4% of the patients presented with fever, 90.5% had localized pain, and only 19% had gastroenteritis. The most common level of discitis was the lumbar region, especially the L4/L5 level. Primarily, third-generation cephalosporin was administered, and antibiotic treatment was given for an average of 9.6 weeks. Non-typhoid *Salmonella* spondylodiscitis is a rare clinical entity in healthy and immunocompetent children. The identification of the responsible organism is essential to guide antibiotic therapy and define the treatment duration. A significant limiting factor in this systematic review was the lack of published research articles and case series due to the rarity of the disease.

## 1. Introduction

*Salmonellae* cause a broad spectrum of clinical manifestations, from gastroenteritis, typhoid fever, and bacteremia to the asymptomatic carrier state [1]. Spinal infections are rare in children and thus not often encountered in community and emergency medicine. Its incidence has been estimated at 2.4 cases per 100,000 population, increasing with age (from 0.3 per 100,000 in the <20 years age group to 6.5 per 100,000 in the over 70 years group [2]). The average age of spinal infection (vertebral osteomyelitis or spondylodiscitis) in children is approximately two to eight years, and the incidence of lumbar or lumbo-sacral region involvement represents the majority of cases, although any spinal level can be affected [3]. A wide range of organisms have been associated with spinal infections. *Staphylococcus aureus* has been identified to cause approximately 80% of the verified spinal infections that occur in the first months of life and in most of those that develop in older children [4]. *Mycobacterium tuberculosis* is also a common cause of spinal infection, especially in developed countries. *Escherichia coli*, *a-hemolytic streptococcus*, *streptococcus pneumoniae*, and *Kingella kingae* are some organisms identified as causatives of spinal infections in children in the literature [4,5,6]. Sporadic case reports of *non-typhoid Salmonella* spinal infection have also been published, most of them in adults [7]. In the pediatric population, few cases of spinal infections caused by *Salmonella non-typhi* have been described, and most affected children presented with predisposing factors such as sickle cell disease, malignancies, and other immunocompromised conditions [8].

Because of the rarity of discitis due to *Salmonella* in immunocompetent children, accurate diagnosis is often delayed or even missed.

The optimal treatment of spondylodiscitis caused by *Salmonella* is controversial. Some authors report that most patients respond well to antibiotics alone, whereas others favor surgical treatment combined with antibiotics [9,10].

This study compiled multiple studies with the aim of summarizing the diagnosis and treatment methods for non-typhoid *Salmonella* spondylodiscitis in children to provide guidance for clinical practice.

## 2. Materials and Methods

### 2.1. Literature Search

The systematic review was conducted based on the preferred reporting items for systematic reviews and meta-analyses (PRISMA) guidelines, including multiple scientific databases (PubMed, EMBASE, and the Cochrane Library). The MeSH terms “spondylodiscitis” and its corresponding synonyms (discitis, spinal osteomyelitis) were designated keywords, and the term “Salmonella” was combined in “AND” form in the search strategy.

In our review, we used only studies with humans. The procedures were performed by two independent reviewers. Any disagreement was resolved by discussion.

### 2.2. Selection Criteria

All the retrieved studies met the following inclusion criteria: (i) the original study topic involved children (<18 years old) with non-typhoid Salmonella spondylodiscitis; (ii) the necessary information of clinical outcomes, diagnosis, and treatment protocols were included, and (iii) the studies were case series, case reports, and prospective or retrospective cohort studies.

Unfortunately, because of the rarity of non-typhoid *Salmonella* spondylodiscitis in healthy children, the studies included in the systematic review are mainly case reports and one case series.

The exclusion criteria were as follows: (i) the study was published in a non-English language or was a non-human study; (ii) the study included a duplicate population.

### 2.3. Quality Assessment

The quality of the eligible studies was evaluated by two authors independently, using different methods according to the study design: the Joanna Briggs Institute (JBI) Critical Appraisal Checklist for Case Reports [11] and the Joanna Briggs Institute (JBI) Critical Appraisal Checklist for Case Series [12].

### 2.4. Data Extraction

The information from the selected studies was reviewed by two independent researchers. Age, sex, medical history, and diagnosis were included. Clinical information included the affected spinal location, systemic and local symptoms and signs, laboratory test results, pathogen types, antibiotic types, administration routes, duration, operations, and treatment outcomes. Successful treatment was based on (i) no symptoms and signs of infection, (ii) no subsequent surgical intervention; and (iii) no death caused by a condition directly linked to non-typhoid *Salmonella* spinal infections in the children.

### 2.5. Statistical Analysis

Statistical analysis was performed using SPSS v29.0 (SPSS, Inc., Chicago, IL, USA). Continuous normally distributed variables were expressed as means and standard deviations and analyzed using Student’s *t*-test. Continuous non-normally distributed variables were expressed as medians and ranges and analyzed with the Mann–Whitney U-test. Categorical variables were expressed as proportions and percentages and analyzed with the chi-square test or Fisher’s exact test (when appropriate). A *p*-value of < 0.05 was considered statistically significant.

## 3. Results

### 3.1. Literature Search Results

Details of the literature search process are shown in the flow chart (Figure 1). In total, 273 publications were obtained from the literature. By excluding duplicates (n = 155) and scanning titles and abstracts (n = 75), 43 articles were enrolled for full text review. We were not able to retrieve two of the articles. Of the 41 studies selected for eligibility, 18 studies were excluded since there was no information available regarding the demographics of the subjects, the exact site of the infection, and the serotype of *Salmonella*. Moreover, three of the articles included spinal infection with *Salmonella typhi* in children. Finally, 20 studies were selected for quality assessment. Information of 21 cases of *non-typhoid Salmonella* spondylodiscitis in children were included in our systematic review (Appendix A, Table A1). As mentioned before, due to the rarity of the clinical condition, only case reports and one case series have been published in the literature (Figure 1).

### 3.2. Demographic Characteristics

The age range of non-typhoid *Salmonella* discitis was variable, ranging from 2 years old to 18 years old. There was no case study in children less than 1 year old. The average age was 12.76 years among the 21 cases of non-typhoid *Salmonella* spondylodiscitis.

Concerning the sex prevalence, 11/21 (52.4%) were female, and 10/21 (47.6%) were male (Table 1).

### 3.3. Clinical Features, Physical Examination, and Comorbidities

The clinical manifestation of spinal infections due to non-typhoid *Salmonella* varied. Most of the children in the literature 19/21 (90.5%) presented with back pain that pre-existed the diagnosis and was localized in the part of the spine affected by the infection. In toddlers (two children), the most frequent reported symptom was limping with refusal to walk, stand, or sit 2/21 (9.5%). Fever was present in 11/21 (52.4%) of the cases at admission.

Extraskeletal manifestations of *Salmonella* infection were reported in only 4/21 (19%) of the children. The predominant extraskeletal symptom was gastroenteritis, which was combined with diarrhea and abdominal pain. In 17 out of 21 cases (81%), no abdominal irritation or gastroenteritis in the months before the diagnosis of non-typhoid *Salmonella* was reported.

None of the patients reported any comorbidities; they were healthy immunocompetent children with no prior admissions to hospital, based on the studies.

The imaging modalities used for the diagnosis of spondylodiscitis were various. Most patients initially underwent standard X-ray imaging, during the emergency department evaluation, with anterior–posterior and lateral projections. In 8/21 (38.09%) computed tomography (CT) of the affected region was performed, and the diagnosis of spondylodiscitis was established [13,14,15,16,17,18,19,20]. An MRI of the affected region was performed in 13/21 of the cases reviewed (61.9%) [21,22,23,24,25,26,27,28,29,30,31,32].

The most affected site of the spine was the lumbar segment, especially L4/L5, with 33.3% (7/21) of cases having localized infection at that level. Next, L1/L2 and L3/L4 had 3/21 (14.3%) for each level. At the thoracic spine, 2/21 (9.5%) of the cases were located at the level T10/T11 (9.5%) and T11/T12 (9.5%). One case was reported for T7/T8 (4.8%), T12/L1 (4.8%), L2/L3 (4.8%), and L5/S1 (4.8%). (Table 2).

### 3.4. Laboratory Investigations

Regarding the laboratory data, the results from all the studies were heterogeneous and could not be merged analytically. Moreover, in some of these studies, laboratory data were not reported, or were reported partially.

In our systematic review, 17 of the 21 cases had detailed laboratory tests. The mean value of the white blood cells (WBC) was 12.153 /10^3^/mm^3^. Moreover, the mean CRP value was 7.7071 mg/dL. Increased values of the inflammatory markers were noticed mainly in the patients who presented with fever.

In all cases, blood cultures were taken; however, only 28.6% of the cases (6/21) were positive for non-typhoid *Salmonella*. In 71.4%, the results were negative for *Salmonella*. The above results show the poor sensitivity of the blood cultures in spinal infections due to non-typhoid *Salmonella*. Stool cultures were sent in all the patients, and 4/21 (19%) had gastroenteritis but only one case was positive for non-typhoid *Salmonella* (4.8%) (Table 2).

The verification of the infection in 16/21 (76.2%) children was performed via CT-guided (computed tomography guided) biopsy/needle aspiration technique, and in 1/21 (4.8%) children it was performed via an open biopsy, which yielded positive results. In some cases where there was a positive blood culture, and the patients were toddlers, the attending physicians decided not to perform biopsies and started treatment based on the blood cultures alone (4/21 cases, 19%) (Table 2).

### 3.5. Etiology and Pathogenesis

All the cases in the systematic review were immunocompetent patients without risk factors. All the children were tested for various immunodeficiencies and relevant underlying conditions, but all the tests were negative.

All infections were monomicrobial, and 11 different serotypes of non-typhoid *Salmonella* were identified. The most common non-typhoid *Salmonella* serotype that caused spinal infection was the *Salmonella enterica* in 6/21 (28.6%) of the affected children. The second most common serotype was the *Salmonella Group B* with 5/21 (23.8%) of the affected children. Moreover, in two cases (9.5%) *Salmonella Oranienburg* was detected. In the remaining cases, *Salmonella Saintpaul*, *Arizonae*, *Lomita*, *Paratyphi*, *Cerro*, *Panama*, *Agona*, and *Salmonella Group C1* were identified in one patient each (4.8%) (Table 3).

### 3.6. Treatment and Clinical Outcomes

Antibiotic treatment alone was utilized in 17 of 21 cases (81%). Debridement of the affected disc (surgical intervention) was performed in four cases (19%) combined with appropriate antibiotic treatment based on the antibiogram of the *non-typhoid Salmonella* serotype. Eventually, 20/21 (95.2%) of the cases were cured and had excellent outcomes; in one case (4.8%) chronic osteomyelitis of the vertebra was reported with occasional back pain in follow-up.

Regarding medical therapy, intravenous ceftriaxone was administrated in 5/21 cases (23.8%) and intravenous ciprofloxacin was utilized in 3/21 cases (14.3%). In 3/21, cases intravenous ceftazidime was given, and intravenous ampicillin was administrated in 4/21 cases (19%). Cefotaxime, aztreonam, piperacillin, pefloxacin, chloramphenicol, and moxalactam were used in one case each (4.8%) intravenously. In three cases (14.3%) the duration of the intravenous antibiotics was 8 weeks. Intravenous administration discontinued in 7/21 (33.3%) of the cases in 6 weeks, in two cases (9.5%) in 7 weeks, and in five patients (23.8%) in 4 weeks. According to the literature, most of the cases continued with oral antibiotics. Four patients (19%) continued treatment with oral trimethoprim-sulfamethoxazole, and four patients (19%) with oral administration of ciprofloxacin. Cotrimoxazole and ampicillin were given in two cases (9.5%). Treatment was continued for an average of 9.61 weeks (95% CI, 6–18 weeks) [13,14,15,16,17,18,19,20,21,22,23,24,25,26,27,28,29,30].

## 4. Discussion

Spondylodiscitis is generally caused by the hematogenous spread of bacteria to the spine and the tissues around it or from the direct spread of purulent lesions [33]. Clinical manifestations of discitis in children include local pain, fever, and neurological symptoms. Local pain is a frequently observed symptom, and severe pain with neurology often suggests an epidural abscess [34].

Spondylodiscitis has been diagnosed with increasing frequency [35]. In the period in which MRI was not available, the incidence of SD was estimated by Digby and Kersley to be approximately 1:250,000 of the population [36]. Some years later, Cushing calculated that cases primarily involving the disc were 1–2 per year per 32,500 pediatric hospital evaluations [37]. In the pediatric population, 1%–2% of all osteomyelitis cases have been reported to begin with spinal involvement.

*Staphylococcus aureus* and *S. epidermidis* account for 60% and *Enterobacteriaceae* for 30% of the causative organisms of infectious discitis. In a review of more than 7000 cases, only 0.76% of patients with *Salmonella* infection had osteomyelitis [38], most commonly involving the diaphyseal part of the long bones such as the humerus and the femur. Spinal infection has been reported in only 0.45% of all *Salmonella* osteomyelitis [9], with the lumbar vertebra being the most common site of involvement [7].

Children with hemoglobinopathies, especially sickle cell disease, are believed to be predisposed to infection with *Salmonella* species, due to repetitive vaso-occlusive crises causing devitalization of the gut and bone, red cell breakdown products of chronic hemolysis saturating the macrophage system, and underlying splenic and hepatic dysfunction. In our systematic review, we included only case reports and case series of non-typhoid *Salmonella* spondylodiscitis in healthy immunocompetent children [39].

The principal reservoirs for non-typhoid *Salmonella* organisms include birds, mammals, reptiles, and amphibians, and the major food vehicles of transmission to humans in industrialized countries include foods of animal origin or food contaminated by an infected animal or human carrier. There are few case reports for non-typhoid *Salmonella* spinal infection in immunocompetent children. In Japan, the most prevalent serotype in human salmonellosis is *Enteritidis*, and it is often associated with contaminated eggs. There were three Japanese case reports of spondylodiscitis caused by *Salmonella* in immunocompetent children [28,29,32], and in one case it was strongly suspected that consumption of a dried squid product was associated with the infection course [28].

Non-typhoid *Salmonella* spondylodiscitis in children does not differ in terms of clinical presentation with the spondylodiscitis caused by the usual organisms. Discitis presents with mild clinical manifestation in children less than 4 years old. Low grade fever and mild pain are the main features of the clinical presentation. Early clinical signs of spinal infection are nonspecific, so the diagnosis of discitis is made late in the course of the disease. Several studies reported delays of 4–6 months after the initiation of the symptoms in establishing a diagnosis [37,40,41]. Older children and adolescents present with high fever, distinguished localized pain, and a clinical picture resembling the one of the spinal infections in adults [42]. Gastroenteritis is a clinical syndrome that may be present in children with spinal infection due to non-typhoid *Salmonella* in a small percentage. Based on our systematic review, only 19% of the reported cases presented with abdominal pain and diarrhea and verified non-typhoid *Salmonella* discitis.

The laboratory characteristics of the patients with non-typhoid *Salmonella* spondylodiscitis were nonspecific. The inflammatory markers such as white blood cells, ESR, and CRP can be normal. Blood cultures were positive for non-typhoid *Salmonella* in 28.6% of the cases with spinal infection. In a recent systematic review of Spondylodiscitis in children, only 13.37% of the cases had positive blood cultures [43].

When spinal infection is suspected, imaging is the key. Plain X-rays of the spine remain within normal limits in most of the cases of discitis even after prolonged disease [44]. A useful tool is a bone scan with technetium for the diagnosis of inflammatory changes in children in spine. Spots of increased radiotracer accumulation highlight inflammatory changes [45,46]. An MRI has been the most reliable method in detecting infectious discitis in children, with high sensitivity and specificity, and it is considered the method of choice for diagnosis of spinal infection. Several MRI patterns and signal intensity alterations have been described to be indicative of infectious discitis [47,48,49].

Despite the sensitivity of the imaging modalities and the technological advances, identification of the organism responsible for the spinal infection is necessary. Due to the invasiveness of a biopsy in children, often, clinicians decide to continue with empirical therapy for common organisms that cause spinal infection in the area of the diagnosis, based on the laboratory results and imaging. CT-guided biopsy or open biopsy are performed in children who fail to improve with empirical antibiotic therapy, when there is suspicion of atypical microorganism or when the lesion of the vertebra resembles a tumor [50]. In cases with positive blood cultures, it was assumed that the organism responsible for the spinal infections was the same. It has been reported that the yield from blood cultures varied between 40% and 60% in clinically defined cases of pyogenic spondylodiscitis [6]. In our systematic review, only 28.6% of the children with discitis due to non-typhoid *Salmonella* had positive blood cultures.

According to the literature, *Staphylococcus aureus* and *Streptococcus species* predominate in the cultures of positive cases of pyogenic spondylodiscitis in children [51]. Immunocompromised individuals and children with sickle cell anemia may develop *Salmonella* discitis. Strains of *Kingella kingae* (a Gram-negative coccobacillus of low virulence) causes self-limiting spondylodiscitis in children under 3 years of age (6 months–4 years old) [5]. Tuberculosis of the spine is a significant health burden in developing countries, though it is also an emerging problem in the developed world as well. In children, discitis is more aggressive that in adults, and they usually present with back pain. The region most commonly affected is the thoracolumbar [52]. In our systematic review, one of the patients with discitis was initially treated with antituberculosis drugs. After 2 weeks of therapy the patient continued to present fever and the same level of pain as in the beginning, and a CT-guided biopsy was performed, which identified non-typhoid *Salmonella* [19].

Establishing the underlying etiology of the spinal infection is crucial, as it determines the duration of treatment and choice of antimicrobial agents used to treat the condition. Antibiotics is the first line of treatment, and operative treatment is only indicated in cases with neurological deficit and spinal instability. Although for adults, official guidelines for spinal infections have been prepared, no guidelines exist for the pediatric population with discitis. When antibiotic therapy is required due to the severity of the clinical presentation in children, the choice of the drug (s) should be based on the sensitivity tests of the causative agent (if available) or the known sensitivity to antibiotics of the pathogens that commonly cause spinal infections in the area where the disease is diagnosed. In almost all cases reported in this systematic review, there was a significant delay in the identification of the non-typhoid *Salmonella* that was responsible for the discitis. According to the studies available, while awaiting laboratory tests, a combination of broad-spectrum antibiotics, including a drug active against *S. aureus*, is given intravenously for 3–4 days [43]. Biopsy was essential to identify the pathogen in 17/21 cases (80.9%) included in the systematic review.

Numerous factors affect the effectiveness of the antibiotic treatment in *Salmonella*’s spondylodiscitis, including the patient’s age, the type of antimicrobial agent, the antibiotic sensitivity, the severity of the disease, the co-existence of immunocompromised conditions, and the duration of diarrhea before the starting of antibiotic treatment if there is gastroenteritis before [44]. Although non-typhoid *Salmonella* spinal infections are extremely rare in the literature, with only 21 cases found, there is the emergence of antibiotic resistance [53,54]. The most common antibiotics used for non-typhoid spinal infection were ampicillin, trimethoprim–sulfamethoxazole, and chloramphenicol. 

The American Academy of Pediatrics currently recommends third-generation cephalosporin for 4–6 weeks extra-intestinal non-typhoid *Salmonella* infections [55,56]. Treatment failures have been reported even after these prolonged courses and may occur days to weeks following completion of therapy and apparent complete clinical and culture response [54]. Relapses are likely recrudescence rather than reinfection and reflect the intracellular persistence of non-typhoid Salmonella and the inadequate penetration of antibiotic into the macrophages [57]. The duration of antibiotics is very important, as an increased rate of recurrence has been described when the duration is insufficient. However, no data on failure and recurrence were available, and most of the included cases that achieved favorable results were treated for 6 to 8 weeks. Based on our systematic review, the mean duration of the antibiotic treatment was 9.6 weeks of intravenous and oral treatment. Based on the reviewed cases, 90.5% of the children received intravenous antibiotic therapy for non-typhoid *Salmonella*, according the antibiogram, for a mean period of 5.6 weeks (minimum 1 week, maximum 8 weeks). In total, 61.9% of the cases received oral antibiotic therapy with a mean duration of 6.4 weeks (minimum 2 weeks, maximum 12 weeks).

The duration of antibiotic treatment should be decided based on clinical features and imaging. Further studies need to be conducted for the optimal duration of treatment. Clinical trials are very difficult to perform given the rarity of pyogenic discitis in children and especially non-typhoid *Salmonella*-induced infections in healthy children.

## 5. Limitations

Our systematic review had several limitations. First, the included studies were all case series and case reports, and the quality of the original studies was limited. Bias may have impacted the results due to the small sample size, and unpublished literature was not included in our study.

## 6. Conclusions

Spondylodiscitis in immunocompetent children is a rare clinical entity, and clinicians should have a high index of suspicion when a child with back pain and fever is presented. Non-typhoid *Salmonella* discitis is easier to diagnose with the use of modern and more efficient diagnostic methods. Identification of the responsible organism is crucial for the appropriate antibiotic treatment. Non-typhoid *Salmonella* spinal infection in healthy children is extremely rare but should be considered as a causative pathogen. Salmonella spondylodiscitis usually responds to appropriate antibiotics. Surgical treatment is necessary only in cases with neurology and spinal instability.

## Figures and Tables

**Figure 1 children-09-01852-f001:**
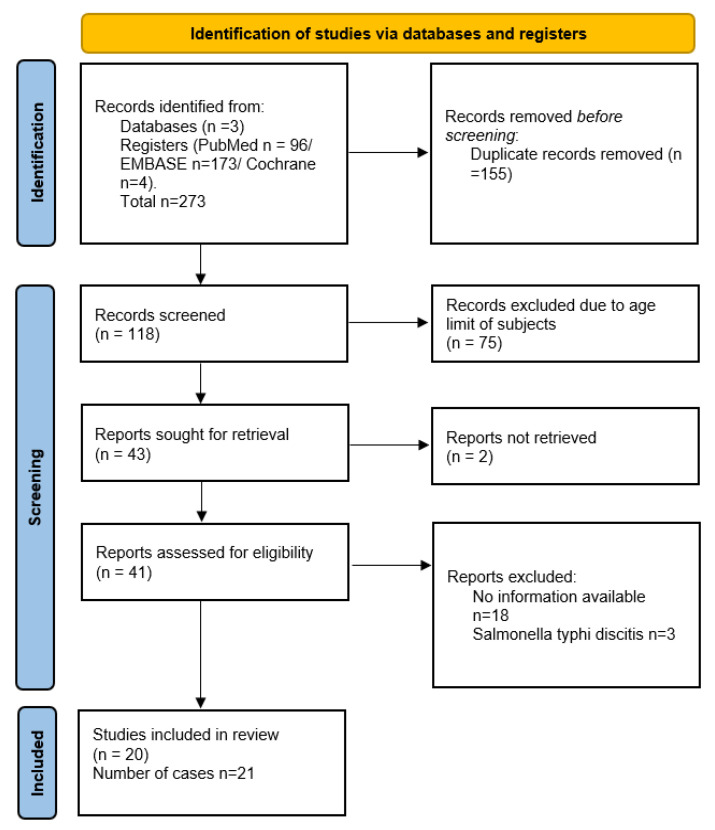
Flow diagram for the systematic review of non-typhoid *Salmonella* spinal infections in immunocompetent children.

**Table 1 children-09-01852-t001:** Clinical features in the cases of NTS spinal infection in healthy children.

Clinical Features in Cases of NTS Spinal Spondylodiscitis	
**Feature**	**Value**
**Age**	
Mean	
Range	12.82–18
**Sex**	
Female	11 (52.4%)
Male	10 (47.6%)
**Medical History**	None
**Symptoms**	
Fever	11 (52.4%)
Localized pain	19 (90.5%)
Gastroenteritis	4 (19%)
**Level**	
Thoracic	6 (28.5%)
Lumbar	15 (71.4%, L4/L5 33.3%)

**Table 2 children-09-01852-t002:** Laboratory investigations in cases of NTS spinal infections in healthy children.

Clinical Test Results in Patients with NTS Spondylodiscitis	
WBC	Mean 12.153 × 10^3^/mm^3^ Range: 6000–19,000 × 10^3^/mm^3^
CRP	Mean 7.7 mg/dL Range: 0.20–60 mg/dL
Blood Cultures	Negative 15/21 (71.4%) Positive 6/21 (28.6%)
Biopsy	None performed 4/21 (19%) CT-Guided Biopsy 16/21 (76.2%) Open Biopsy 1/21 (4.8%)

**Table 3 children-09-01852-t003:** Serotypes Identified for NTS Spinal Infections in Children.

	Frequency	Percentage	Valid Percentage	Cumulative Percentage
*SALMONELLA ENTERICA*	6	28.6	28.6	28.6
*SALMONELLA SAINTPAUL*	1	4.8	4.8	33.3
*SALMONELLA GROUP B*	5	23.8	23.8	57.1
*SALMONELLA ARIZONAE*	1	4.8	4.8	61.9
*SALMONELLA LOMITA*	1	4.8	4.8	66.7
*SALMONELLA GROUP C1*	1	4.8	4.8	71.4
*SALMONELLA PARATYPHI*	1	4.8	4.8	76.2
*SALMONELLA CERRO*	1	4.8	4.8	81.0
*SALMONELLA PANAMA*	1	4.8	4.8	85.7
*SALMONELLA ORANIENBURG*	2	9.5	9.5	95.2
*SALMONELLA AGONA*	1	4.8	4.8	100.0
Total	21	100.0	100.0	

## Data Availability

Not applicable.

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
