# Peer review of "A Systematic Review of the Diagnosis and Treatment of Non-Typhoid Salmonella Spondylodiscitis in Immunocompetent Children"

_children, 2022, doi:10.3390/children9121852_

Round 1

Reviewer 1 Report

Abstract

Well written and clearly describes the study.  Concise description of the methods, results and limitations.

Introduction

---Would italicize Staphylococcus aureus, and use a lower case a for ‘aureus’

---For the second sentence, it is not clear if you are referring to Salmonella spinal infections or spinal infections in general.  Would clarify.

---Would italicize all genus/species names

Methods

---Good use of multiple databases

---Appreciate the reliance on PRISMA guidelines

---Was ‘human’ used as a search filter, or were articles screened manually for human relevance after identification?

---Would note that the title and aim of the study (to evaluate Salmonella spinal infections) is not fully accomplished with the search terms used.  For example, ‘epidural abscess’ is a MeSH term which may be considered a spinal infection but was not included. Nor was ‘osteomyelitis’ in general, with manual searching for cases of vertebral osteomyelitis which may have been missed by the MeSH terms used.  Would recommend addressing this in the manuscript, and/or changing the title and aim of the study.

---For section ii of 2.2, what was considered ‘necessary information’ in this setting? 

---Section 2.4.  Diagnosis is misspelled as disgnosis

---Section 2.5 would include more details as to the types of statistical testing used (e.g. Chi-square test, t-test)

Results

---Figure 1 reads well

---Section 3.2.  Not sure you need 2 significant digits for age (12.76 years; would round to 12.8 years)

---Table 2:  would label age as ‘years’ for units.  Would also consider making the sub-headings of Age, Symptoms, Gender, etc stand out:  as it currently reads, they look the same as the variables underneath them.

---What type of imaging modality was used in the patients (MRI vs plain film)?

Discussion

---Line 258, page 7:  In one third of which cases?  There is no reference provided

---Lines 296-299:  if spinal infections are so rare, how can you be sure there is emergence of antibiotic resistance in these infections?  Or are you referring to Salmonella infections in general?

---Line 299:  not sure chloramphenicol is used that often anymore for treatment. 

---Good discussion of relapses and the pathophysiology of this

---Under limitations, I would include missed cases based upon the search terms used.

Appendix A, table 1:  How were outcomes classified as ‘good’ or ‘excellent’?  What criteria were used?

References

Reference 6---the journal is Chemotherapy not Chemoter

Reference 7:  Proper abbreviation would be Clin Infect Dis

Author Response

Reviewer 1

Abstract

Well written and clearly describes the study.  Concise description of the methods, results and limitations.

 Introduction

Point 1: Would italicize Staphylococcus aureus, and use a lower case a for ‘aureus’

Response 1: Changed according to reviewer’s suggestion.

Point 2: For the second sentence, it is not clear if you are referring to Salmonella spinal infections or spinal infections in general.  Would clarify.

Response 2: The words “in general” is added to clarify the meaning of the sentence.

Point 3: Would italicize all genus/species names

Response 3: Changed according to reviewer’s suggestion.

 Methods

---Good use of multiple databases

---Appreciate the reliance on PRISMA guidelines

Point 4: Was ‘human’ used as a search filter, or were articles screened manually for human relevance after identification?

Response 4: All articles were screened manually for “human” as a search filter initially and then manually after identification for verification.

Point 5: Would note that the title and aim of the study (to evaluate Salmonella spinal infections) is not fully accomplished with the search terms used.  For example, ‘epidural abscess’ is a MeSH term which may be considered a spinal infection but was not included. Nor was ‘osteomyelitis’ in general, with manual searching for cases of vertebral osteomyelitis which may have been missed by the MeSH terms used.  Would recommend addressing this in the manuscript, and/or changing the title and aim of the study.

Response 5: All relevant sections in the manuscript and the title were modified. With the modifications the title is in accordance with the aim of the study.  

Point 6: For section ii of 2.2, what was considered ‘necessary information’ in this setting

Response 6: The necessary information was considered sufficient when all of the below were included in the study:

  1. History
  2. Detailed clinical  examination
  3. Laboratory test results (WBC, ESR, CRP, blood tests)
  4. Methods of diagnosis of spondylodiscitis
  5. Methods of identification of responsible microorganism (open biopsy, CT guided biopsy, blood cultures)
  6. Treatment (surgical or conservative)
  7. Antibiotic treatment and duration of therapy
  8. Symptoms and clinical examination after the completion of therapy

Point 7: Section 2.4.  Diagnosis is misspelled as disgnosis

Response 7: Corrected.

Point 8: Section 2.5 would include more details as to the types of statistical testing used (e.g. Chi-square test, t-test)

Response 8: More details regarding statistical analysis were added.

 Results

---Figure 1 reads well

Point 9: Section 3.2.  Not sure you need 2 significant digits for age (12.76 years; would round to 12.8 years)

Response 9: Changed.

Point 10: Table 2:  would label age as ‘years’ for units.  Would also consider making the sub-headings of Age, Symptoms, Gender, etc stand out:  as it currently reads, they look the same as the variables underneath them.

Response 10: Table 2 was revised according to reviewer’s suggestions.

Point 11: What type of imaging modality was used in the patients (MRI vs plain film)?

Response 11: Various imaging modalities were used. More details were added in the text (lines 143-148).

Discussion

Point 12: Line 258, page 7:  In one third of which cases?  There is no reference provided

Response 12: The sentence was modified.

Point 13: Lines 296-299:  if spinal infections are so rare, how can you be sure there is emergence of antibiotic resistance in these infections?  Or are you referring to Salmonella infections in general?

Response 13: We are referring to infections caused by Salmonella in general. The relative references that support this statement are provided (ref 55, 56).

Point 14: Line 299:  not sure chloramphenicol is used that often anymore for treatment

Response 14:  The use of chloramphenicol among other antibacterial drugs, is suggested as treatment option for azithromycin-resistant Salmonella serotypes in the paper by Robert-Jan H et al. (reference 57). The relative reference was added in the text.

Chloramphenicol was one of the antibiotics used for treatment of Salmonella spondylodiscitis based on the case studies included in our review [18,19,21]. Nowadays, chloramphenicol is not used for the treatment of Salmonella spondylodiscitis in children.

---Good discussion of relapses and the pathophysiology of this

Point 15: Under limitations, I would include missed cases based upon the search terms used.

Response 15: The sentence “Due to the rarity of non-typhoid Salmonella spondylodiscitis, there is a possibility of missed cases based upon the search terms used” was added in the Limitations section.

 Point 16: Appendix A, table 1:  How were outcomes classified as ‘good’ or ‘excellent’?  What criteria were used?

Response 16: The outcomes were classified as Good and Excellent based on the MacNab Criteria:

  1. Excellent: No pain, full activity with work
  2. Good: Occasional pain, not interfering with work
  3. Fair: Occasional pain, interfering with work
  4. Poor: Pain persists, frequently interferes with work

References

Point 17: Reference 6---the journal is Chemotherapy not Chemoter

Response 17: Corrected.

Point 18: Reference 7:  Proper abbreviation would be Clin Infect Dis

Response 18: Corrected.

Reviewer 2 Report

The analyses themselves are sound and I believe the results of their work. The data are well organized by the authors. I, therefore, recommend this paper be published in the Children Journal after the authors address the following comments.

·         Review English grammar as there are mistakes throughout the text. This article should be completely rewritten.

·         An abstract is not well organized. The abstract must be improved. The authors must explain the application and novelty of the research work add in the abstract section.

·         The literature section must be improved with more advanced articles and clearly why your present study is different, better to explain novelty.

Author Response

Reviewer 2

The analyses themselves are sound and I believe the results of their work. The data are well organized by the authors. I, therefore, recommend this paper be published in the Children Journal after the authors address the following comments.

 Point 1: Review English grammar as there are mistakes throughout the text. This article should be completely rewritten.

Response 1: A paid editing service (MDPI) was used to correct English grammar mistakes.

Point 2: An abstract is not well organized. The abstract must be improved. The authors must explain the application and novelty of the research work add in the abstract section.

Response 2: Abstract was revised according to reviewer’s suggestions.

Point 3: The literature section must be improved with more advanced articles and clearly why your present study is different, better to explain novelty.

Response 3: To our knowledge, our study is the first that summarizes the existing literature on diagnosis and treatment of non-typhoid Salmonella spondylodiscitis. Due to the rarity of the disease the relative studies are scarce and no advanced articles exist in the literature. We clearly explain all the above in the literature section.
